# Enhanced Heterogeneous Peroxymonosulfate Activation by MOF-Derived Magnetic Carbonaceous Nanocomposite for Phenol Degradation

**DOI:** 10.3390/ma16093325

**Published:** 2023-04-24

**Authors:** Xinyu Li, Xinfeng Zhu, Junfeng Wu, Hongbin Gao, Weichun Yang, Xiaoxian Hu

**Affiliations:** 1Chinese National Engineering Research Center for Control & Treatment of Heavy Metal Pollution, Institute of Environmental Engineering, School of Metallurgy and Environment, Central South University, Changsha 410083, China; 2Henan Key Laboratory of Water Pollution Control and Rehabilitation, Henan University of Urban Construction, Pingdingshan 467000, China

**Keywords:** Cu-Co@C, metal-organic framework, peroxymonosulfate, phenol, sulfate radical

## Abstract

Degradation efficiency and catalyst stability are crucial issues in the control of organic compounds in wastewater by advanced oxidation processes (AOPs). However, it is difficult for catalysts used in AOPs to have both high catalytic activity and high stability. Combined with the excellent activity of cobalt/copper oxides and the good stability of carbon, highly dispersed cobalt-oxide and copper-oxide nanoparticles embedded in carbon-matrix composites (Co-Cu@C) were prepared for the catalytic activation of peroxymonosulfate (PMS). The catalysts exhibited a stable structure and excellent performance for complete phenol degradation (20 mg L^−1^) within 5 min in the Cu-Co@C-5/PMS system, as well as low metal-ion-leaching rates and great reusability. Moreover, a quenching test and an EPR analysis revealed that ·OH, O_2_·^−^, and ^1^O_2_ were generated in the Co-Cu@C/PMS system for phenol degradation. The possible mechanism for the radical and non-radical pathways in the activation of the PMS by the Co-Cu@C was proposed. The present study provides a new strategy with which to construct heterostructures for environmentally friendly and efficient PMS-activation catalysts.

## 1. Introduction

The use of sulfate radicals based on advanced oxidation processes (SR-AOPs) has been considered as an attractive and viable approach to decreasing the persistence of organic compounds in water [1]. These SR-AOPs can be more effective at degrading persistent organic compounds due to the advantages they offer, including higher oxidation potential, higher selectivity, higher pollutant selectivity, and longer half-lives [2]. Phenol is a representative and persistent organic pollutant [3]. It is also a model pollutant for evaluating advanced oxidation performance [4]. Therefore, we chose phenol as the target contaminant. There are many form of persulfate activation, such as thermal, microwave, ultrasonic, catalyst-based, etc. In particular, transition-metal-based catalysts are widely used as PMS activators due to their low toxicity, high abundance, and high catalytic activity [5]. Transition-metal oxides, such as Cu_x_O_y_, Mn_x_O_y_ etc., have been proven to effectively activate peroxymonosulfate (PMS) [6]. For example, the Mn_3_O_4_/CuBi_2_O_4_ composite material designed by Zhang et al. [7], used for the catalytic activation of PMS, exhibits high phenol (almost 100%) and TOC (74.3%) degradation rates within 10 min. However, the direct exposure of these catalysts in bulk solutions inevitably leads to metal leaching, resulting in secondary contamination. 

The combination of transition-metal oxides with carbonaceous materials is not only conductive to the elimination of metal leaching, but also improves catalytic activity [8]. Firstly, this kind of composite can be tailored with unique structures (e.g., core-shell, capsule structure) to reduce direct the exposure of metals [9]. Secondly, the introduced carbon materials possess active sites and oxygen functional groups that can be involved in the catalytic activation of PMS, and can serve as a matrix to disperse the transition-metal-oxide particles to prevent agglomeration [10]. Li et al. [11] demonstrated that the combination of transition metals and carbon can regulate electronic states. Jiang et al. [12] prepared highly nitrogen-doped copper-oxide-nanocomposite carbon materials (CuO-NC), which were able to remove 4-chlorophenol completely (30 min), compared to undoped copper-oxide nanocomposites. Although much progress has been made, ensuring that catalysts have both superior reusability and high catalytic activity remains a challenge. 

In our previous study, we developed Cu-Cu_x_O composite carbon nanomaterials and used them to degrade organic pollutants, and the performance was restored through the calcination of the reacted Cu-Cu_x_O@C, which was conductive to the reuse of the catalysts. However, there problems remained, such as difficulty of separating the catalysts from the solution, gradually decreasing the catalytic performance and gradually deteriorating the stability [13]. Compounds based on Co (e.g., Co_3_O_4_) have demonstrated excellent activity for oxidation reaction due to its strong redox performance. Furthermore, since they exhibited good magnetic characteristics, they can easily provide magnetic separation [14]. 

In this study, we conceived and prepared a composite structure of cobalt-oxide and copper-oxide nanoparticles embedded in a carbon matrix (Co-Cu@C) via a polyvinylpyrrolidone (PVP)-assisted-growth-pyrolysis method [15]. The Co-Cu@C exhibited enhanced catalytic activity and excellent reusability for phenol degradation, with a removal efficiency of almost 100% (20 mg L^−1^ phenol), which was maintained for 5 min and reused three times. The mechanism through which the Co-Cu@C activated peroxymonosulfate, involving radical and non-radical pathways, was revealed. The MOF-derived carbon-complex-synthesis strategy constructed in this study has significant potential for the catalytic degradation of organic compounds in wastewater.

## 2. Materials and Methods

### 2.1. Materials

Parabenzoquinone(p-BQ), 1,3,5-Benzenetricaiboxylic acid (BTC), and oxone (KHSO_5_ ≥ 42.8%) were bought from Shanghai Macklin Biochemical Technology company. Cobalt nitrate (Co(NO_3_)_2_·6H_2_O), cupric nitrate (Cu(NO_3_)_2_·3H_2_O), polyvinylpyrrolidone (PVP), phenol (PhOH), peroxymonosulfate (PMS, 2KHSO_5_·KHSO_4_·K_2_SO_4_), methanol (MeOH, CH_4_O), tert-butanol (TBA, C_4_H_10_O), furfuryl alcohol (FFA), H_2_SO_4_, and NaOH were obtained from Sinopharm Chemical Reagent company (Shanghai, China).

### 2.2. Synthesis of Cu-CuxO@C

The method of preparation of the material was based on the relevant research and was improved [13,16]. First, the Co-doped HKUST-1 composites were prepared by hydrothermal method. A specific ratio of Cu(NO_3_)_2_·3H_2_O and Co(NO_3_)_2_·6H_2_O was mixed in 20 mL deionized water. Next, 500 mg of polyvinylpyrrolidone was dissolved in the aqueous solution. The molar ratios of Cu(NO_3_)_2_·3H_2_O to Co(NO_3_)_2_·6H_2_O were 5:1, 4:2, 3:3, 2:4, and 1:5, respectively. The BTC (4 mmol) was dispersed in 20 mL of ethanol. The solutions described above were poured together into the Teflon liner and mixed well, and then the Teflon line with mixed solution was loaded into a sealed stainless-steel autoclave and heated at 120 °C for 24 h. Next, the sediments were separated by centrifuging, cleaned four times with MeOH, and dried at 120 °C for 12 h. The materials with different copper-to-cobalt ratios of 5:1, 4:2, 3:3, 2:4, and 1:5 were synthesized and recorded as HKUST-1/Co-x (x = 1, 2, 3, 4, 5). The synthesized HKUST-1/Co powders were poured into alumina crucibles and placed in tube furnaces at the calcinated rate of 5 °C min^−1^ and calcined in nitrogen (N_2_) atmosphere at 700 °C for 2 h. The black powders were obtained after cooling to ambient temperature in N_2_ atmosphere. The prepared powders with various ratios were named Cu-Co@C-x (x = 1, 2, 3, 4, 5).

### 2.3. Characterization of the Synthesized Activators

The morphologies of the synthesized activators were ascertained by high-resolution transmission-electron microscopy (HR-TEM, FEI TF20, Hillsboro, OR, USA) equipped with energy-dispersive spectroscopy (EDS, Super-X). Raman-spectra patterns were recorded with excitation wavelength of 514 nm by reflex Raman spectroscopy (Renishaw, Wotton-under-Edge, UK). Further experimental details are given in Appendix A.

### 2.4. Catalytic Experiments

The Cu-Co@C catalytic performance was examined through the degradation of phenol. All degradations were tested by batch experiments with the 100-mL beakers based on the magnetic stirrers. Metal leaching was determined by ICP-OES (Agilent 5100, Santa Clara, CA, USA) for Cu^2+^ and Co^2+^ at wavelengths of 324 nm and 235 nm. Electron paramagnetic resonance (EPR, Bruker EMXplus-6/1, Billerica, MA, USA) was used for experiments on reactive oxygen species (ROS). Further experimental details are given in Appendix A.

## 3. Discussion

### 3.1. Structure and Morphology Characterization

We used TEM and EDS mapping to compare the morphological and element distribution of the Cu-Co@C. The particle scales of the Cu-Co@C with different Cu-Co ratios were about 2~5 µm (Figure 1). This indicated that the structures of the Cu-Co@C particles did not collapse with the introduction of the cobalt element and that there was no significant effect on the particle size. It was observed from the EDS mapping images (Figure 2) that the C element was mainly distributed in the outer layer of the particles, and the distribution of the Co element was generally the same as that of the C element. Xu et al. [17] demonstrated that Co can catalyze the formation of graphite from carbon-containing precursors, which are wrapped in the outer layer of cobalt. In this study, cobalt-oxide particles were embedded in the carbon matrix. Thus, the Co element in the Cu-Co@C was uniformly distributed in the carbon structure. In addition, in combination with Figure 1 and Figure 2, it can be seen that the copper oxide in the Cu-Co@C, except for that in the Cu-Co@C-5, overflowed from the surface of the particles. During the pyrolysis process, the graphite structure formed in the Cu-Co@C-5 effectively prevented the internal copper oxide from expanding outward. The encapsulated structure effectively reduced the leaching of metals in water.

Various Cu–Co ratios appeared in the XRD patterns of the Cu-Co@C (Figure 3). The peaks varied slightly across the different Cu-Co@C samples. All the Cu-Co@C samples displayed three diffraction peaks at 43.29°, 50.42°, and 74.18°, which were indexed as PDF no. 04–0836 to the typical (111), (200), and (220) phases of the Cu^0^, respectively [13,18]. Moreover, the (400) plane of the Co_3_O_4_ peak located at 44.8° (PDF no. 42–1467) [19] was gradually enhanced as the molar ratio of the Co increased. It was noteworthy that the diffraction peaks of the cobalt oxides were weak in intensity and broad, which indicated that the Co_3_O_4_ or CoO crystals in Cu-Co@C were mainly nanometer-sized microcrystals [20]. Interestingly, the characteristic peak of cobalt oxide decreased with increasing cobalt content, while a diffraction peak at 26.1° in the Cu-Co@C-5 was attributed to the (002) plane of graphite [21]. Combined with the HRTEM images shown in Figure 4a, it can be observed that the cobalt-oxide particles were dispersed in the carbon shell of the Cu-Co@C-5. The hierarchical graphite structures were generated around the cobalt-oxide particles. This was in agreement with the XRD analysis. Specifically, due to the wrapping effect of the graphite structure and the small particle size, the characteristic peaks of the cobalt oxide were not detected completely. In addition to the (400) plane of the Co_3_O_4_ (d = 0.220 nm), the (111) plane of the Co (d = 0.204 nm) and the (111) plane of CoO (d = 0.245 nm) were observed by HRTEM (Figure 4b) [22,23]. Different samples of Co with various valence states have excellent electron-transfer effects as catalysts [24,25].

A N_2_ adsorption–desorption isotherm measurement was performed to compare the mesoporous structure parameters in the Cu-Co@C with various Cu–Co ratios (Figure 5a and Appendix A). The isotherms for all the Cu-Co@C were of the typical type II, which revealed that the N_2_ was absorbed at multiple layers on the surfaces of the materials and that the adsorption was reversible [13,20,26]. The BET-specific surfaces (S_BETs_) and pore volumes of the Cu-Co@C-x composites are presented in Appendix A. The S_BETs_ of the Cu-Co@C-x were 112.49, 109.05, 121.09, 175.76, and 189.45 m^2^ g^−1^, respectively. This demonstrated that the SSA of the Cu-Co@C-5 was larger than those of the other Cu–Co ratios. Moreover, the BJH pore-size distribution of the Cu-Co@C is demonstrated in the inset in Figure 5a. The BJH average pore diameters of the Cu-Co@C-x were 11.59 nm, 13.31 nm, 10.96 nm, 8.69 nm, and 7.75 nm, respectively. The pore-size-distribution diagram exhibited that the Cu-Co@C-5 contained both mesopores and macropores. Such structures can not only expose more active sites in solutions [13], but can also enhance the hydrophilicity of materials [26].

Raman spectroscopy was employed to evaluate the loading effect of the heteroatoms and the presence of carbon. Two peaks of carbon, at 1343 cm^−1^ (D-peak) and 1594 cm^−1^ (G-peak), were observed in all the as-prepared Cu-Co@C composites. The degree of disorder in the carbon layer was reflected by the D-peak. The stronger the D-peak, the more defects were present in carbon material. The G-band reflects the crystallinity of sp^2^ hybrid carbon atoms and is used to characterize the degree of graphitization in carbon materials [27,28]. The D and G bands were present in all the tested samples. The values of I_D_/I_G_ for the Cu-Co@C-x were 0.64, 0.48, 0.37, 0.27, and 0.12, respectively. This suggested that the degree of graphitization of the composites increased. It further proved that cobalt is helpful in catalyzing the transformation of disordered carbon structures into graphite structures [29], which was in agreement with the HRTEM and XRD results presented above. In addition, the wettability of the catalyst is critical to catalytic performance. The diffusion resistance of the reactants is reduced through a hydrophilic catalyst [27]. The water-contact angles of the Cu-Co@C-x are shown in Appendix A. The Cu-Co@C with different Cu–Co ratios exhibited good hydrophilicity. The Cu-Co@C-5 showed better hydrophilicity with a water-contact angle of 42.9°. The hydrophilicity of the Cu-Co@C-5 was enhanced by the large number of oxygen-containing groups in the outer graphitic structure of the Cu-Co@C-5.

### 3.2. Catalytic Performances

The material used for the phenol degradation was tested in a solid–liquid heterogeneous catalytic system (Figure 6). The dosage of Cu-Co@C was 0.5 g L^−1^, the dosage of PMS was 2 mmol L^−1^, and the pristine solution of phenol was 20 mg L^−1^. Firstly, adsorption tests (30 min) on the phenol materials were carried out to avoid the effects of adsorption in the catalytic process. As shown in Appendix A, the maximum adsorption efficiency of the Cu-Co@C-x (x = 1, 2, 3, 4, 5) for the phenol was 12.2%, 20.8%, 24.6%, 46.5%, and 52.7%, respectively. In addition, it was difficult to degrade the phenol with 2 mmol L^−1^ of PMS without a catalyst (Appendix A). The degradation of the phenol took place in a ternary system of catalyst–PMS–phenol. In the catalytic system in which the Cu-Co@C and PMS coexisted, the concentration of phenol was significantly reduced. In particular, the phenol degraded completely in the Cu-Co@C-5/PMS system. The significant increase in phenol removal was due to the ROS generated by the Cu-Co@C/PMS system. Interestingly, except for the Cu-Co@C-5-PMS system, all the other tests showed a decrease and then an increase in the phenol concentration in the catalytic reaction process. On one hand, this might have been due to the positive charge accumulated on the surface of the material during the PMS activation, which would have encouraged the adsorption of phenol. As the activity of the material decreased, phenol was desorbed in the solution. On the other hand, combined with the characterization results described above, the spilled copper-oxide particles were leached in the catalytic process, resulting in material-structure destruction and phenol desorption into the solution [19]. The catalytic activities compared are listed in Appendix A.

A series of conditioned experiments (activator dosage, PMS dosage, and pH conditions) were used to assess the phenol-degradation performance of the Cu-Co@C-5 (Figure 7). Moreover, adsorption tests (30 min) on the phenol materials were carried out to avoid the effects of adsorption in the catalytic process. At room temperature, further SR–AOP studies were carried out with different catalyst dosages (0.3, 0.5, and 0.75 g L^−1^) and a PMS dosage of 2 mmol L^−1^. During the adsorption phase, the adsorption of the phenol increased with increasing amounts of catalyst (Figure 7a). At the catalytic reaction stage with the PMS, during the tests with different catalyst dosages, the phenol was completely removed within 3 min. The effects of various PMS concentrations on the degradation performance were also tested. As shown in Figure 7b, except for the PMS system with 0.5 mmol L^−1^, all the PMS systems completely degraded the phenol within 3 min. This was because higher concentrations of PMS can provide more radicals for phenol degradation [28,30]. In general, the activation of PMS for the degradation of organic contaminants is greatly influenced by the pH of the solution. Interestingly, in the Cu-Co@C-5/PMS system, the performance of PMS activated via the Cu-Co@C-5 for phenol degradation was almost unaffected by the pH value. Under operating conditions of pH 3.0–11.0, the phenol was completely degraded within 10 min and the adsorption performance of the material was not reduced. This indicated that Cu-Co@C-5 has a broad pH range in which phenol degradation can take place with activated PMS. In addition, the pattern of pH changes during the reaction was studied. Notably, the pH of the solution after the reaction was always maintained at about 5.4 with an initial pH of 3.0–11.0 (Appendix A). This indicated that the Cu-Co@C-5 possessed excellent solution-buffering ability during the catalytic process. Similar results were found in our previous studies [31,32].

Stability experiments were performed on the Cu-Co@C-5. The Cu-Co@C-5 was tested for three cycles to investigate the durability of the catalyst (Figure 8a). The efficiency of the Cu-Co@C-5’s activation of the PMS to degrade phenol reached 100% in the three cycles. This suggested that the Cu-Co@C-5 retained its excellent catalytic activity and high stability during the reuse processes. When comparing the XRD patterns of the Cu-Co@C-5 before and after the catalytic reaction, the crystalline-phase stability of the Cu-Co@C-5 was established (Figure 8b). This further proved that Cu-Co@C-5 possesses excellent stability. In addition, in the adsorption stages of the three cycles without PMS, the Cu-Co@C-5’s capacity to adsorb phenol within 30 min was measured at 53.7%, 7.4%, and 7.3%, respectively. In the 2nd and 3rd cycles, the phenol-adsorbing capacity of the Cu-Co@C-5 decreased significantly due to the phenol-degradation products adsorbed on the material surface [33,34]. The catalytic performance of the Cu-Co@C-5 was not affected by the reduction in the adsorbing capacity, which further proved that the phenol removal was due to degradation via the Cu-Co@C-5-activated PMS rather than by adsorption. In addition, in the process of activating the PMS, the Cu-Co@C-5 also showed superior ionic stability (Figure 8c). The rates of Cu^2+^ leaching from the Cu-Co@C-x (x = 1, 2, 3, 4, 5) were 117.6, 103.0, 99.0, 83.8, respectively, and 16.5 mg L^−1^, and those of the Co^2+^ leaching from the Cu-Co@C-x were 12.9, 16.3, 24.3, 26.7, and 4.3 mg L^−1^, respectively. It appears that the ion leaching of the Cu-Co@C-5 was obviously lower than those of the other tested materials, which may have been due to the graphite-carbon coating on the composite surface inhibiting the overflow of metal ions, effectively reducing the dissolution of the metals in the reaction process. The metal particles dissolved in contact with the solution because the pH of the reaction system was weakly acidic. The metal particles in the composites from the Cu-Co@C-1 to the Cu-Co@C-4 overflowed from the carbon support, which increased the likelihood that they would interact with the solution. However, the presence of a graphitized carbon layer in Cu-Co@C-5 can effectively reduce metal-oxide dissolution. The magnetic properties of the Cu-Co@C-5 at room temperature were evaluated by a vibrating sample magnetometer (VSM), and the magnetic hysteresis loop is shown in Figure 9. The Cu-Co@C-5 exhibits superparamagnetic properties [35,36]. In a magnetic field of 20,000 G, the saturation-magnetization strength of the Cu-Co@C-5 was about 78 emu g^−1^. The strong magnetic properties of the Cu-Co@C-5 ensured that it was attracted to external magnetic fields in the solution (Figure 9). Its easy separation facilitated the recycling of the material in the water-treatment process. Further cost–benefit-analysis details are given in Appendix A.

### 3.3. Identification of Radicals

Typically, the degradation of the organics was accomplished by activating the free radicals generated by the PMS. The activation of PMS may produce a variety of reactive radicals. The free-radical species involved in the phenol degradation in the Cu-Co@C-5/PMS system were determined by using radical-quenching experiments. We used MeOH, TBA, FFA, and BQ as free-radical-probe chemicals. In previous studies, MeOH was reported to rapidly trap ·OH (9.8 × 10^8^ M^−1^ s^−1^) and SO_4_·^−^ (1.6 × 10^7^ M^−1^ s^−1^), while TBA only trapped ·OH (4.1 × 10^8^ M^−1^ s^−1^) [13,23,37]; FFA and BQ were used to quench O_2_^·−^ (kO_2_·^−^,BQ = 5.0 × 10^8^ M^−1^ s^−1^) [19,24] and ^1^O_2_ (k_1_O_2_,FFA = 5.0 × 10^8^ M^−1^ s^−1^) [38,39], respectively. To depict the effects of different scavenger agents in the degradation reaction, the degradation data of the phenol were described by pseudo-first-order kinetics (Appendix A). The liner first-order relationship was shown by plotting the ln(C/C_0_) vs. the reaction time.

Unlike the MeOH and TBA, the FFA and BQ both inhibited the reaction, suggesting that the oxidative degradation of the phenol was related to the ·OH, O_2_·^−^, and ^1^O_2_ (Figure 10a). Although the phenol was completely degraded in the reaction system using MeOH and TBA as quenching agents, the kobs of the reaction decreased from 0.4156 min^−1^ to 0.2523 min^−1^. This indicates that a small amount of ·OH participated in the oxidative degradation of the phenol. In the FFA/PMS and BQ/PMS systems, the degradation was significantly inhibited. The degradation rates over 20 min were 28.2% and 44.8%, and the kobs of the reaction were 0.0035 min^−1^ and 0.0060 min^−1^, respectively. The FFA had the most obvious inhibitory effect on the phenol degradation, indicating that the ^1^O_2_ played an important role in the reaction, followed by the O_2_·^−^. Thus, the ordering of the degradation of phenol by free radicals in the system was as follows: ^1^O_2_ > O_2_·^−^ > ·OH > SO_4_·^−^.

The electron paramagnetic resonance (EPR) spectra further confirmed the production of free radicals in the Cu-Co@C-5/PMS system. We used DMPO and TEMPO to capture the free radicals. A representative signal of DMPO-·OH is aN = aH = 14.9 G [28,33]. As shown in Figure 10b, only ·OH was detected in the Cu-Co@C-5/PMS/DMPO system. This was in agreement with the results of the radical-quenching tests, further confirming that there was no SO_4_·^−^ in the reaction system. After quenching the ·OH and SO_4_·^−^ with methanol as the solvent, the presence of O_2_·^−^ was detected in the Cu-Co@C-5/PMS/DMPO system (Figure 10c). The Cu-Co@C-5/PMS/TEMPO system also displayed a large amount of ^1^O_2_, which was generated in the catalytic process (Figure 10d). In addition, compared with the strength of the free radicals after 2 min and 10 min of reaction, the strengths of the ·OH and ^1^O_2_ significantly increased, suggesting that the free-radical production was sustained during the reaction. The slight decrease in the intensity of the O_2_·^−^ at 10 min was due to the fact that part of the O_2_·^−^ was consumed to generate the ^1^O_2_, while the other part was consumed as it participated in the phenol degradation.

### 3.4. Cu-Co@C-5 Activated PMS and Phenol-Degradation Mechanisms

#### 3.4.1. Reaction Pathway of Cu-Co@C-5

To determine the changes in the front elements and their valence states in the catalyst before and after the degradation reaction, the Cu-Co@C-5 was analyzed by XPS (Figure 11 and Appendix A). The proportions of carbon, cobalt, copper, and oxygen in the fresh Cu-Co@C-5 were 81.30%, 7.80%, 2.81%, and 8.09%, respectively. The Co and Cu were mainly in the lower valence states of Co^0^, Co^2+^, Cu^0^, and Cu^+^. According to the fitting analysis of the Co-element-binding energy, the contents of Co0, Co^2+^, and Co^3+^ were 55.8%, 31.0%, and 13.2% respectively. The contents of Cu^0^, Cu^+^, and Cu^2+^ were 59.2%, 13.6%, and 27.2%, respectively, according to the Auger spectra analysis of the Cu elements. The percentages of C, Co, Cu, and O after the reaction were 66.33%, 9.70%, 0.60%, and 23.38%, respectively. The contents of the Co and Cu in the three valence states changed significantly, mainly in the form of high valence states. The contents of the Co^3+^ and Co^2+^ increased from 31.0% and 13.2% to 39.2% and 41.8%, respectively, and the contents of the Cu^2+^ and Cu^+^ increased from 27.2% and 13.6% to 39.7% and 20.2%, respectively. The contents of the zero-valence metal elements decreased. This indicated that the metal ions underwent redox during the catalytic reaction process, and that free radicals were generated in the process of electron transformation [13,17,19,32,40]. In addition, the contents of C=O and C-OH also changed. The electron transfer between the Cu-Co@C-5 and the peroxymonosulfate was facilitated by the C-O with high redox potential. The C-OH bond played the role of the electron-donating group and was also used as the active site in the catalytic reaction. These bonds exert a synergetic effect on catalytic activated PMS in phenol-degradation processes [13,28].

A comparison of the XPS of the Cu-Co@C-5 before and after the reaction and the results reported in previous research enables speculation as to the cyclic pathways of the metal-valence states. The standard reduction potential of Co^3+^/Co^2+^ (E^0^ = 1.92 eV) is much higher than that of Cu^2+^/Cu^+^ (E^0^ = 0.34 eV). Therefore, Co^3+^ can be reduced to Co^2+^ by Cu^+^ (Equation (1)). Conversely, in the transition from Co^2+^ to Co^3+^, electrons are lost, which can reduce Cu^2+^ to Cu^+^ (Equation (2)) [41,42]. In addition, the relative contents of Cu^0^ and Co^0^ decreased after the reaction, indicating that they were oxidized, which also provided Cu^+^ and Co^2+^ for the cycling of the metal-valence state (Equation (3)) [43,44,45,46,47]. The redox abilities of the Cu-Co@C-5 and the monometallic materials were compared through a cyclic voltammetry (CV) analysis (Appendix A). The Cu-Co@C-5 exhibited a higher current density and a stronger reduction ability in the coordination of the redox process [48]. On the CV test, a significant redox reaction occurred on the surface of the Cu-Co@C-5.
(1)$${Cu}^{+}+{Co}^{3+}\to {Co}^{2+}+{Cu}^{2+}$$
(2)$${Co}^{2+}+{Cu}^{2+}\to {Co}^{3+}+{Cu}^{+}$$
(3)$${Co}^{0}/{Cu}^{0}\overset{{2e}^{-}}{\to }{Co}^{2+}/{2Cu}^{+}$$

#### 3.4.2. Electron-Transfer Pathways

The free-radical-quenching experiment showed that the ^1^O_2_ and O_2_·^−^were involved in the degradation of the phenol. However, the HSO_5_^−^ in the PMS was unable to produce any free radicals. Combining the results of the analysis in Section 3.3 and Section 3.4.1, the generation mechanism of the main active radicals in the Co@C-5/PMS system was proposed. The degradation of the phenol involved both free-non-radical and -radical pathways. Firstly, the HSO_5_^−^ obtained electrons to generate ·OH (Equation (4)) [49]. Furthermore, the HSO_5_^−^ hydrolysis produced H_2_O_2_ (Equation (5)). Next, HO_2_·^−^ was produced by the H_2_O_2_ and ·OH through Equation (6). Subsequently, the HO_2_·^−^ decomposed to produce O_2_·^−^ (Equation (7)). The O_2_·^−^ further reacted with water to form ^1^O_2_ (Equation (8)) [40]. In addition, more lattice oxygen was released from the highly dispersed transition-metal-oxide nanoparticles [19,36,50]. The lattice oxygen on the surface of the Cu-Co@C-5 (Osurface) activated the PMS to produce ^1^O_2_ via electron transfer (Equation (9)) [40,49]. Finally, the ·OH, O_2_·^−^, and ^1^O_2_ oxidized the phenol to H_2_O and CO_2_ or intermediate products (Equation (10)).
(4)$${\mathrm{HSO}}_{5}^{-}+e^{-}\to \cdot \mathrm{OH}+{\mathrm{SO}}_{4}^{2-}$$
(5)$${\mathrm{HSO}}_{5}^{-}+H_{2}O\to H_{2}O_{2}+{\mathrm{HSO}}_{4}^{-}$$
(6)$$\cdot \mathrm{OH}+H_{2}O_{2}\to {\mathrm{HO}}_{2}^{\cdot -}$$
(7)$${\mathrm{HO}}_{2}^{\cdot -}\to H^{+}+O_{2}^{\cdot -}$$
(8)2O2·−+2H2O→O21+H2O2+OH−
(9)Osurface+HSO5−→O21+HSO4−
(10)·OH+O2·−+O21+Phenol→H2O+Production

## 4. Conclusions 

In this work, we constructed a novel, encapsulated, and structured Co-Cu@C catalyst with highly dispersed copper-oxide and cobalt-oxide nanoparticles within the carbon structure. The Co-Cu@C-5 displayed outstanding catalytic properties for PMS activation in phenol degradation. The efficiency with which the phenol (20 mg L^−1^) was degraded reached 100% within 5 min, and the removal efficiency remained at 100% after three reuses. The strong PMS activation and reusability of the Cu-Co@C-5 can be attributed to the synergy of the cobalt-oxide and copper-oxide nanoparticles and the carbon matrix. Both free radicals (·OH, O_2_·^−^) and non-free radicals (^1^O_2_) contributed to the decomposition of phenol in the Cu-Co@C-5/PMS during the reaction. It is believed that Cu-Co@C is a promising catalyst for environmental purification.

## Figures and Tables

**Figure 1 materials-16-03325-f001:**
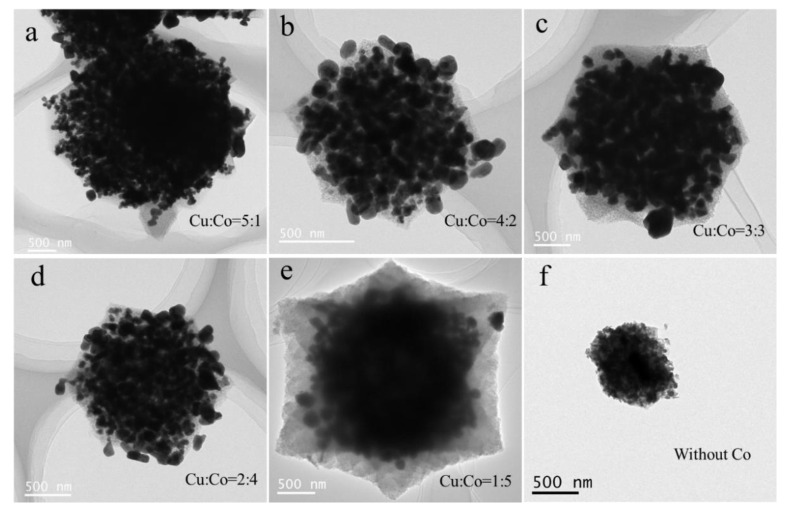
TEM images of Cu-Co@C with various Cu–Co ratios. (**a**) Cu-Co@C-1; (**b**) Cu-Co@C-2; (**c**) Cu-Co@C-3; (**d**) Cu-Co@C-4; (**e**) Cu-Co@C-5; (**f**) Cu_x_O@C.

**Figure 2 materials-16-03325-f002:**
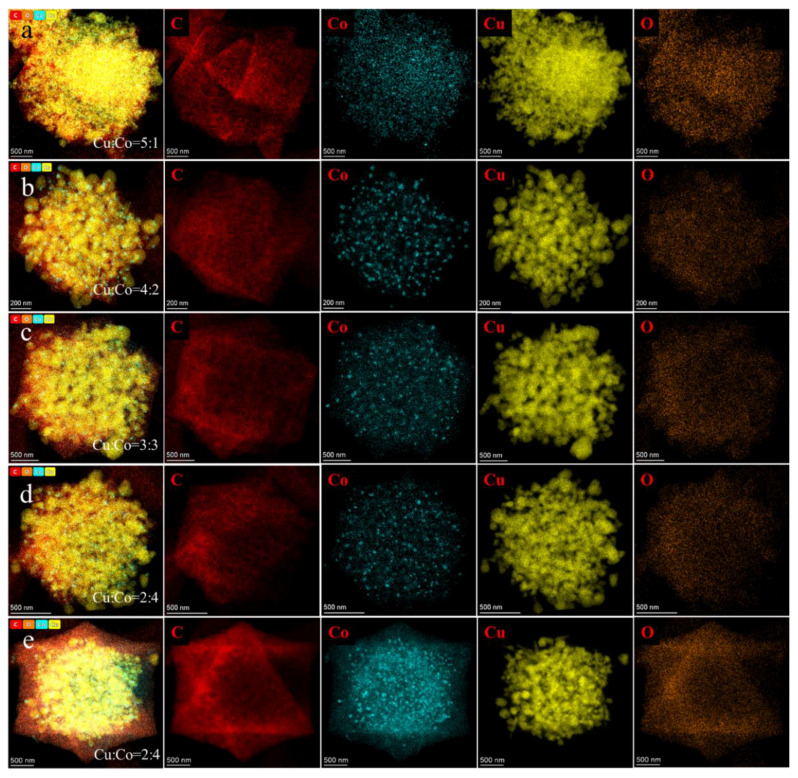
The EDS-mapping images of Cu-Co@C-x with various Cu–Co ratios. (**a**) Cu-Co@C-1; (**b**) Cu-Co@C-2; (**c**) Cu-Co@C-3; (**d**) Cu-Co@C-4; (**e**) Cu-Co@C-5.

**Figure 3 materials-16-03325-f003:**
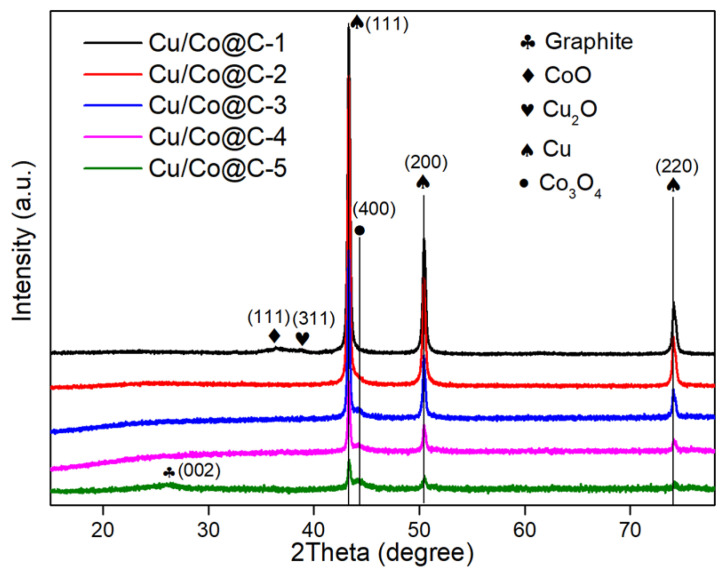
The XRD of Cu-Co@C with various Cu–Co ratios.

**Figure 4 materials-16-03325-f004:**
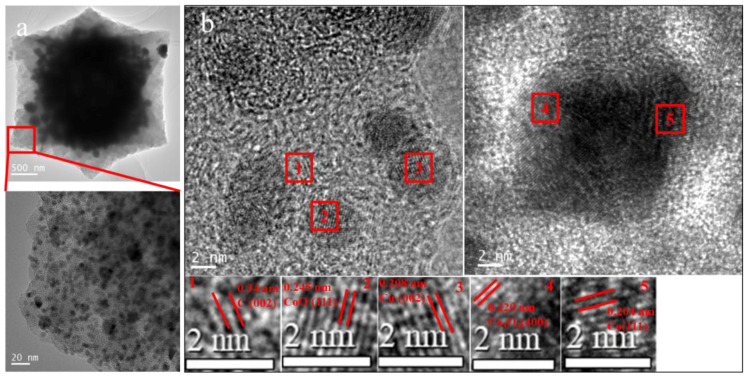
The HRTEM images of Cu-Co@C-5. (**a**) Enlarge image in the selected area; (**b**) Crystal plane analysis of Cu-Co@C-5.

**Figure 5 materials-16-03325-f005:**
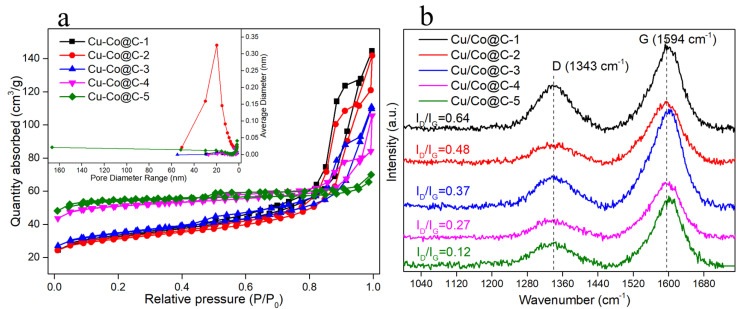
(**a**) N_2_ adsorption–desorption isotherms (inset: pore-size distribution of the Cu-Co@C-x) and (**b**) Raman spectra of as-prepared Cu-Co@C composites with various Cu–Co ratios.

**Figure 6 materials-16-03325-f006:**
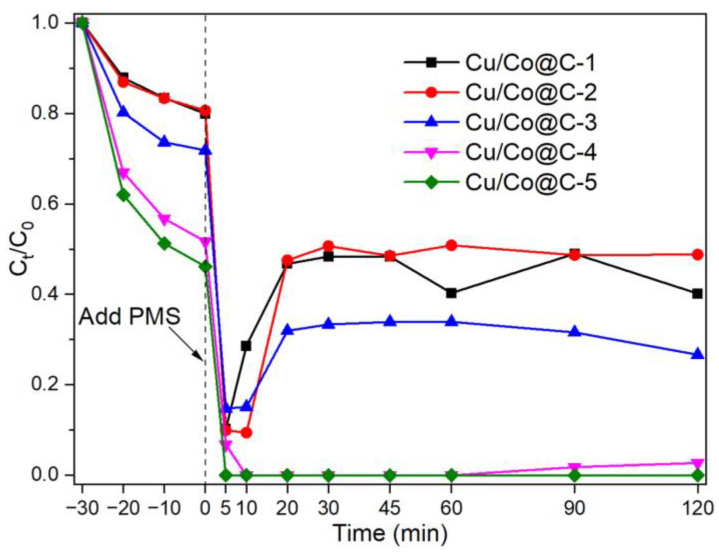
PhOH removal in the presence of Cu-Co@C with various Cu–Co ratios. Experimental conditions: [PMS] = 2 mmol L^−1^, [activator] = 0.5 g L^−1^, [PhOH]_0_ = 20 mg L^−1^.

**Figure 7 materials-16-03325-f007:**
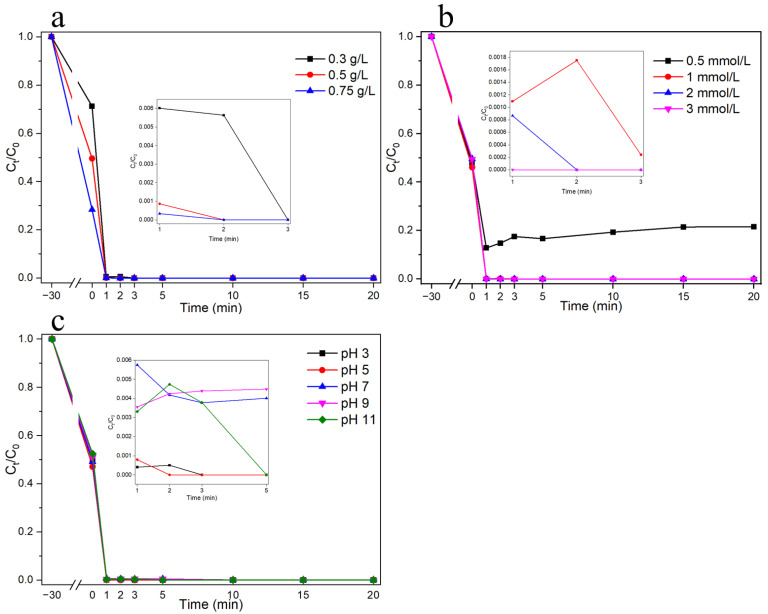
The effects of catalyst dosage ([PhOH]_0_ = 20 mg L^−1^, [PMS] = 2 mmol L^−1^, pH = 6.0) (**a**), PMS dosage ([PhOH]_0_ = 20 mg L^−1^, [Cu-Co@C-5] = 0.5 g L^−1^, pH = 6.0) (**b**), and pH (PhOH]_0_ = 20 mg L^−1^, [PMS] = 2 mmol L^−1^, [Cu-Co@C-5] = 0.5 g L^−1^) (**c**) on phenol degradation in the presence of Cu-Co@C-5.

**Figure 8 materials-16-03325-f008:**
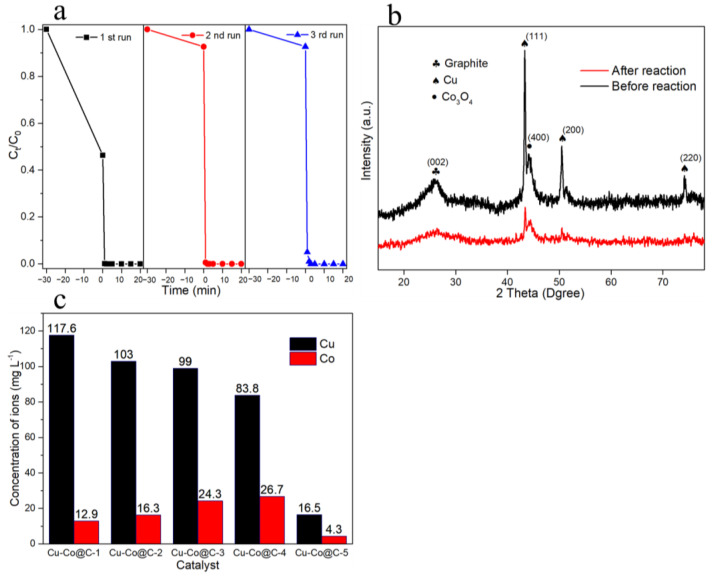
(**a**) Stability measurement of Cu-Co@C-5. (**b**) The XRD patterns after reaction. (**c**) The leaching of Cu and Co from Cu-Co@C-5 (experimental conditions: [PMS] = 2 mmol L^−1^, [activator] = 0.5 g L^−1^, [PhOH]_0_ = 20 mg L^−1^).

**Figure 9 materials-16-03325-f009:**
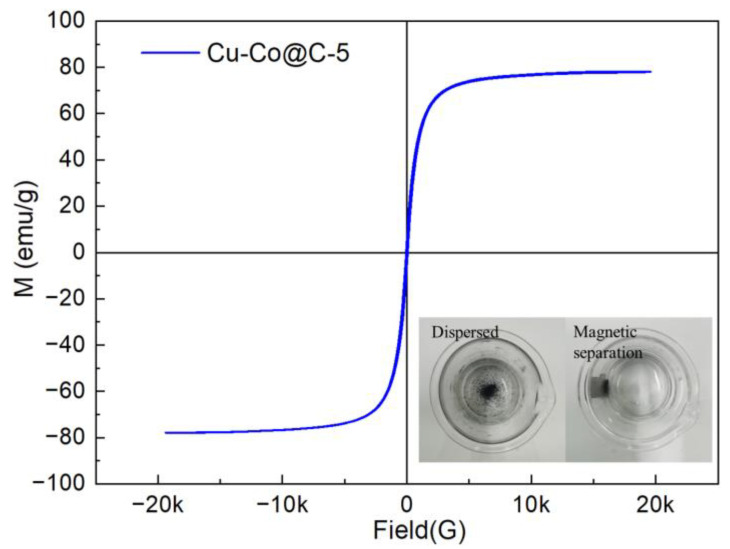
Magnetic hysteresis loops of Cu-Co@C-5 at room temperature (inset: disperser/separation property of Cu-Co@C-5 with external magnetic field).

**Figure 10 materials-16-03325-f010:**
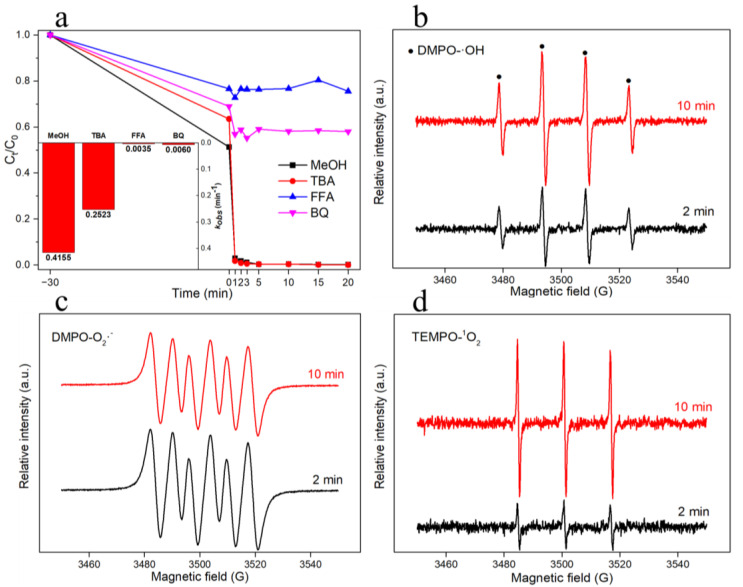
(**a**) Effects of scavengers MeOH, TBA, FFA, and BQ on phenol removal in Cu-Co@C-5/PMS/phenol system (experimental conditions: [PMS] = 2 mmol L^−1^, [activator] = 0.5 g L^−1^, [PhOH]_0_ = 20 mg L^−1^, pH = 6.0, scavenger =1:50). Inset: the effect of the degradation-rate constant for phenol in MeOH, TBA, FFA, and BQ on the pseudo-first-order-rate constants. EPR spectra of DMPO-·OH (**b**), DMPO-O_2_^·−^ (**c**), and TEMPO-^1^O_2_ (**d**).

**Figure 11 materials-16-03325-f011:**
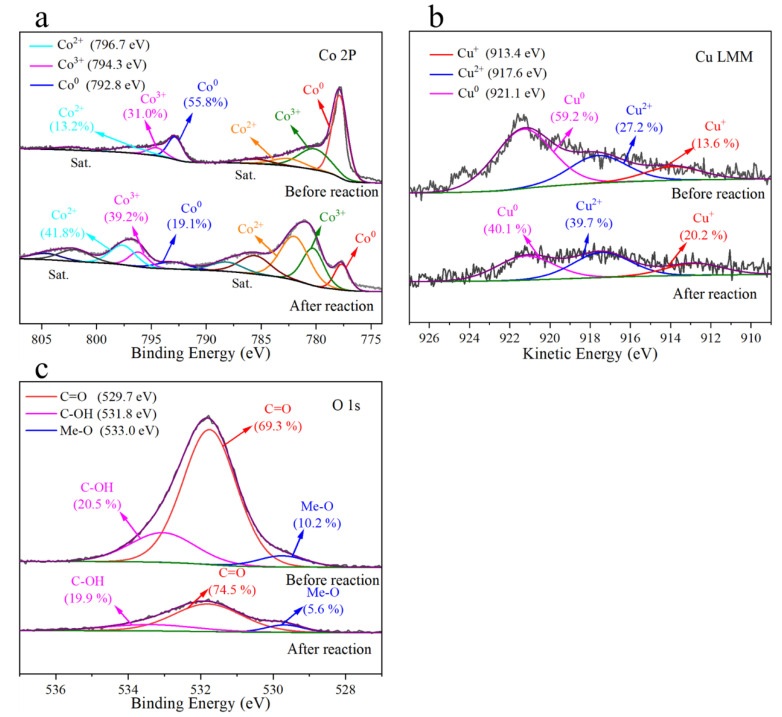
Co 2p spectrum (**a**), Cu LMM spectrum (**b**), and O 1s spectrum (**c**). XPS spectra of the pristine and post-reaction Cu-Co@C-5.

## Data Availability

The data presented in this study are available on request from the corresponding author.

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
