# Peer review of "Enhanced Heterogeneous Peroxymonosulfate Activation by MOF-Derived Magnetic Carbonaceous Nanocomposite for Phenol Degradation"

_materials, 2023, doi:10.3390/ma16093325_

Round 1

Reviewer 1 Report

 The manuscript describes the synthesis of heterogeneous MOFs-derived magnetic carbonaceous nanocomposite and their phenol degradation studies. The design of the experiments and the results presented seem correct and in agreement with each other, their presentation is good, and the manuscript can be considered for publication in its present form. In the opinion of the reviewer, some minor changes are necessary to reconsider the manuscript for publication.

1.       Polyvinylpyrrolidone (PVP)-assisting growth pyrolysis method is the new one or reported already. Include the reference in introduction

2.       “Co-doped HKUST-1 composites” procedure reference could be included

3.       Typographical errors may be corrected eg., 1-day, Co3O4 …etc.,

4.       The dosage of Cu-Co@C was 0.5 g L-1 , the dosage of PMS was 2 mmol L-1 , and the pristine solution of phenol was 20 mg L-1 . Is this the conditions are optimized early ?

5.       Catalytic activities comparison literature may be included to understand the importance of the material. 

Based on the manuscript, the english language writting is good. 

Reviewer 2 Report

Currently many reserchers are looking for solutions to purify contaminated water. However, it is necessary to find solutions that are actually applicable in reality.

This work must be implemented:

- it is necessary to explain well why it was decided to use phenol as a target

- it is necessary to carry out a cost-benefit analysis to verify the effective feasibility

Currently many reserchers are looking for solutions to purify contaminated water. However, it is necessary to find solutions that are actually applicable in reality.

This work must be implemented:

- it is necessary to explain well why it was decided to use phenol as a target

- it is necessary to carry out a cost-benefit analysis to verify the effective feasibility

Reviewer 3 Report

While phenol degradation has been extensively studied using various metal nanoparticles and carbon-supported metal catalysts, the article titled " Enhanced heterogeneous peroxymonosulfate activation by MOFs-derived magnetic carbonaceous nanocomposite for phenol degradation " presents a novel Cu-Co@C-x perspective on this topic. The analytical characterizations, catalytic performance, and proposed mechanisms presented in this scientific article are intriguing. However, to enhance the clarity and rigor of their research, the authors should provide a more detailed explanation for why these nanoparticles are leaching in a separated paragraph before accepting this article. 

The following comments should be addressed.

1. To support the hydrophilicity of all Cu-Co@C-x materials, the authors should provide contact angle measurements which will determine the hydrophobicity/hydrophilicity of the materials. This will aid in understanding the reasons for the high catalytic performance and leaching of nanoparticles. 

2. The analytical characterizations presented in this research are comprehensive and commendable. However, given that most of the catalysts are leaching over the cycles, which is detrimental to heterogeneous catalysis, the authors should focus on minimizing this issue. One potential solution is to reduce the synthesis temperature from 700°C to a lower range (400-500°C).

3. The magnetic properties of Cu-Co@C-x are intriguing, although not entirely unexpected. In order to further explore their potential, it is strongly suggested that the authors perform magnetic susceptibility measurements.

4. Additionally, conducting CV experiments will aid in understanding the reduction potential and catalytic activity for phenol degradation. 

5. To enhance the credibility of their research, the authors should provide NMR data on the phenol degradation production and HRMS data of the catalytic products. These additional analytical measurements will further elucidate the mechanisms underlying the catalytic process.  

Round 2

Reviewer 2 Report

ACCEPT